# Mutation of regulatory phosphorylation sites in PFKFB2 does not affect the anti-fibrotic effect of metformin in the kidney

**Geoff Harley**[1]*, **Marina Katerelos**[1], **Kurt Gleich**[1], **Mardiana Lee**[1], **Peter F. Mount**[1,2], **David A. Power**[1,2]

1 Kidney Laboratory, Department of Nephrology, Austin Health, Heidelberg, VIC, Australia, 2 Department of Medicine, The University of Melbourne, Heidelberg, VIC, Australia

* geoff.harley@austin.org.au

**Data Availability Statement:** All relevant data are within the paper and its Supporting Information files.

## Abstract

The anti-fibrotic effect of metformin has been widely demonstrated. Fibrosis in the kidney after injury is associated with reduced expression of genes involved in both fatty acid and glycolytic energy metabolism. We have previously reported that the anti-fibrotic effect of metformin requires phosphoregulation of fatty acid oxidation by AMP-activated protein kinase (AMPK). To determine whether metformin also acts via regulation of glycolysis, we mutated regulatory phosphosites in the PFKFB2 isoform of 6-phosphofructo-2-kinase/fructose-2,6-biphosphatase (PFKFB2), a key regulator of glycolysis in the kidney. Mice with inactivating knockin (KI) mutations of the phosphorylation sites in PFKFB2 (PFKFB2 KI mice), which reduces the ability to increase the rate of glycolysis following stimulation, were used. Metformin was administered via drinking water to mice with a unilateral ureteric obstruction (UUO) model of renal fibrosis. In the PFKFB2 KI mice treated with metformin, there was decreased fibrosis and macrophage infiltration following UUO as assessed by Western blot for fibronectin and RT-PCR for α-smooth muscle actin, collagen 3, and F4.80, and confirmed by histology. Expression of the inducible PFKFB3 isoform was increased with metformin in UUO in both WT and PFKFB2 KI mice. There was no significant difference between WT and PFKFB2 KI mice treated with metformin in the degree of fibrosis following UUO in any of the Western blot or RT-PCR parameters that were measured. These data show that inhibition of the regulation of glycolysis by PFKFB2 does not diminish the anti-fibrotic effect of metformin in a model of renal fibrosis.

## Introduction

The development of renal fibrosis in response to kidney injury is affected by reduced energy generation in tubular epithelial cells [1]. Fatty acid oxidation is the major source of energy in proximal tubular cells, although glycolysis becomes more important in the low oxygen environment of the medulla and distal nephron. Expression of genes involved in fatty acid oxidation in proximal tubules is reduced in human and experimental renal fibrosis [1].

**Funding:** G. H. was supported by a postgraduate scholarship from the University of Melbourne. D.P. received a National Health and Medical Research Council (NHMRC) grant. There was no additional external funding received for this study.

**Competing interests:** The authors have declared that no competing interests exist.

Administration of drugs such as fenofibrate or metformin, which increase fatty acid oxidation, reduce renal fibrosis [2, 3]. We have reported that, in the case of metformin, the anti-fibrotic effect requires phosphorylation of acetyl-CoA-carboxylase (ACC), which controls entry of cytoplasmic long-chain fatty acids into mitochondria for subsequent β-oxidation [2]. Metformin is an indirect activator of the upstream protein kinase AMP-activated protein kinase (AMPK), which then phosphorylates ACC, leading to reduced fatty acid synthesis and increased fatty acid oxidation in mitochondria.

Interestingly, expression of glycolytic pathway genes is also reduced in proximal tubular cells in renal fibrosis [1]. There are three rate-limiting steps in the glycolytic conversion of a six-carbon ring to two 3-carbon pyruvate molecules, namely hexokinase, phosphofructokinase (PFK) and pyruvate kinase. The rate of glycolysis in cells is directly linked to the rate of glucose entry into the cells as well as the activity of these rate-limiting glycolytic enzymes [4].

Regulation of the rate of glycolysis is predominantly via modification of the activity of PFK. The intracellular level of PFK can be regulated by several factors, but one of the most important is the intracellular level of fructose-2,6-bisphosphate (Fru-2,6-$P_2$), a product of the bifunctional enzyme 6-phosphofructo-2-kinase/fructose-2,6-biphosphatase (PFK-2/FBPase-2), which exists as four isoforms designated PFKFB1-4 [5]. In the kidney, PFKFB2 is described as the predominant isoform [6]. Activity of PFKFB2 is increased by C-terminal domain phosphorylation at Ser[466] and Ser[483]. We have previously shown that inactivating mutations of these PFKFB2 phosphorylation sites, which is predicted to reduce the intracellular level of Fru-2,6-$P_2$, reduced the ability to increase glycolysis in renal tubular cells [4].

Metformin is well established as having protective effects against renal fibrosis in mouse models of disease including ischaemia-reperfusion-injury, unilateral ureteric obstruction (UUO), folic acid nephropathy and cisplatin nephrotoxicity [2, 7–9]. In our previous studies, we have demonstrated that metformin acts, at least in part, by increasing AMPK activation and phosphorylation of ACC, thus stimulating mitochondrial fatty acid oxidation [2]. AMPK is also known to phosphorylate PFKFB2 at its C-terminal end, thereby increasing its activity [5]. In this study, we attempted to determine whether the anti-fibrotic effect of metformin was also dependent on regulation of glycolysis.

## Methods

### Generation of PFKFB2 KI mice

PFKFB2 KI mice with inactivating mutations of the phosphorylation sites of Ser[466] and Ser[483] were generated on a C57Bl/6 background by OzGene Pty Ltd, Bentley DC, WA, Australia as previously described [4]. In brief, phosphor-acceptor sites located in exon 15 of the mouse PFK2 gene were mutated to alanine, generating an inactivating mutation. Mouse genotypes were confirmed by PCR.

PFKFB2 KI mice were maintained as a homozygous line on a C57BL/6 background, whereas WT mice were derived from mating of heterozygous PFKFB2 KI mice and maintained as a separate line. As previously reported, there was no difference in plasma glucose or mouse weight between PFKFB2 KI mice and controls, however, the PFKFB2 KI kidneys were smaller and plasma urea was significantly less [4]. In histological studies, there was no abnormality seen in the PFKFB2 KI kidneys [4]. Cultured tubular epithelial cells from PFKFB2 KI mice have impaired glycolysis when analysis on the Seahorse analyser [4].

### Animal renal fibrosis model

All experiments received prior approval from the Austin Health Animal Ethics Committee which operates under guidelines prepared by the National Health and Medical Research

Council (NHMRC), the Commonwealth Scientific and Industrial Research Organisation (CSIRO) and Animal Welfare Victoria. Since this was an entirely animal study, the requirement for consent was waived. The unilateral ureteric obstruction (UUO) model was used to create renal fibrosis as previously described [10]. In brief, this involved taking male C57Bl/6 mice who were 8–10 weeks old and tying off one of their ureters via a standard surgical technique under isofluorane anaesthesia. Mice were monitored twice daily for weight loss, other signs of distress or evidence of poor health. They received buprenorphine subcutaneous injections for pain in the first three days post-operatively. Seven days later, the mice were sacrificed and a nephrectomy performed. This was done under ketamine anaesthesia with a sufficient dose to provide euthanasia. The obstructed kidney was then used for analysis. In one arm of the study metformin was also added to the mice's drinking water for three days prior to and during the experiment. Liquid metformin hydrochloride (Focus Pharmaceuticals, London, UK) 0.08mg/mL was added into the drinking water and changed every 48 hours.

## Cell culture

Primary cultures of renal tubular epithelial cells (TECs) were prepared by sieving whole mouse kidneys, as we have previously reported [2]. For cell culture stimulation, 4mM metformin (Focus Pharmaceuticals, London, UK) in serum-free media was added to the cells for a period of four hours immediately prior to harvesting. This was based on the experimental design used by Li et al. [7].

## Western blot analysis

Kidney lysate preparation and Western blot analysis was performed via standard methods as previously described [11]. Western Blots were analyzed for densitometry using Image J software. The following antibodies were used in the Western Blot analysis: anti-phospho-PFKFB2 $Ser^{483}$ antibody (Rabbit monoclonal antibody, Cell Signalling Technology, Massachusetts, USA), anti-phospho-ACC $Ser^{79}$ antibody (ABCAM, Cambridge, UK), anti-hexokinase-1 antibody (Rabbit antibody, Cell Signalling Technology, Massachusetts, USA), anti-PFK1M antibody (Rabbit, Sigma-Aldrich, St Louis, USA), anti-α-Smooth muscle actin–FITC antibody (Sigma Aldrich, St. Louis, USA), anti-Fibronectin antibody (Rabbit monoclonal antibody, Sigma-Aldrich, St Louis, USA), anti-PKM2 antibody (Rabbit monoclonal antibody, Cell Signalling technology, Massachusetts, USA), anti-rabbit Immunoglobulin HRP-linked antibody (Swine polyclonal, Dako, Agilent Pathology Solutions, Santa Clara, USA), anti-Fluorescein-POD Fab fragments (Goat, Roche Applied Science, Indianapolis, USA). Anti-GAPDH (Rabbit monoclonal antibody, Cell Signalling Technology, Massachusetts, USA) was used as a loading control.

## Histology

Kidneys were sliced in half transversely and fixed in formalin. Masson's trichrome staining was performed by the Department of Anatomic Pathology, Austin Health. The percentage area occupied by collagen for Masson's trichrome-stained sections was measured using Image J software. This quantitative analysis, performed from a set of images that had coverage of the whole renal cortex, was analysed and the values obtained. The amount of fibrosis weas expressed as a percentage of the whole cortical area.

## Real-time polymerase chain reaction (qRT-PCR)

Total RNA was purified from whole mouse kidney samples and reverse transcribed for analysis as previously described [11]. This was performed using a Stratagen MX3000 real-time PCR

system with Solis Biodyne EvaGreen master mix. The delta-delta Ct method was used to calculate relative expression [12]. Data was expressed as fold expression relative to littermate WT controls.

## Statistics

Statistical analyses were performed using Prism version 7.0a for Mac OS X (GraphPad Software, San Diego, CA). Data are presented as mean + SD. Multiple group means were compared by two-way ANOVA followed by a post-hoc test. Comparison of means from two groups was performed by unpaired t-test. P values of <0.05 were considered significant.

## Results

### Phosphorylation of PFKFB2 and ACC following metformin stimulation in PFKFB2 KI cells

TECs cultured from WT and PFKFB2 KI mice were analyzed via Western blot for expression of phosphorylated PFKFB2 and ACC (Fig 1). As expected, expression of phosphorylated PFKFB2-Ser[483] was not detectable in PFKFB2 KI mice (Fig 1A). Expression of phosphorylation at Ser[466], the phosphorylation target for AMPK on PFKFB2, cannot be distinguished from a similar site in PFKFB3 using existing antibodies due to sequence overlap, so was not examined. AMPK phosphorylates PFKFB2 at its Ser[466] site rather than Ser[483] [5], so it was expected that metformin stimulation would not cause a significant increase in phosphorylation of that site, as shown (Fig 1A and 1B). Metformin stimulation of the TEC's caused a significant

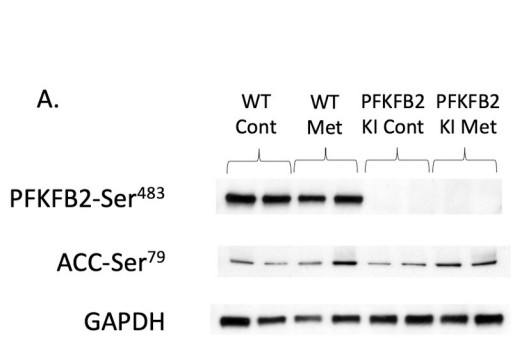

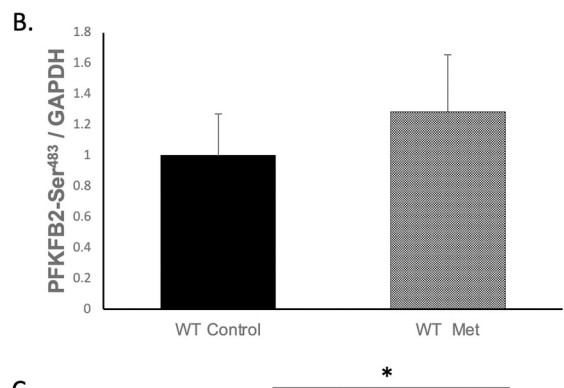

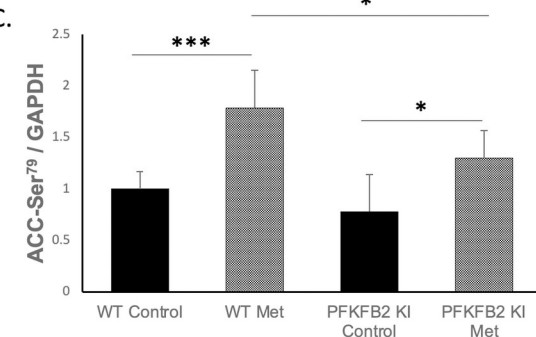

**Fig 1.** Expression of phosphorylation sites in cultured tubular epithelial cells from WT and PFKFB2 KI mice ± metformin stimulation (A-C). The mutated phosphorylation site at PFK-FB2-Ser[483] was not detectable in the PFKFB2 KI mice as expected (**A**). Expression of phospho-Ser[483] was not significantly different with metformin stimulation (**B**). There was increased expression of phosphorylated ACC-Ser[79] phosphorylation with metformin stimulation in both WT and PFKFB2 KI cells (**A, C** ***p<0.001 and *p = 0.01 respectively). Expression of phosphorylated ACC-Ser[79] was decreased in PFKFB2 KI metformin cells compared to WT cells (**A, C** *p = 0.02). Mean + SD.

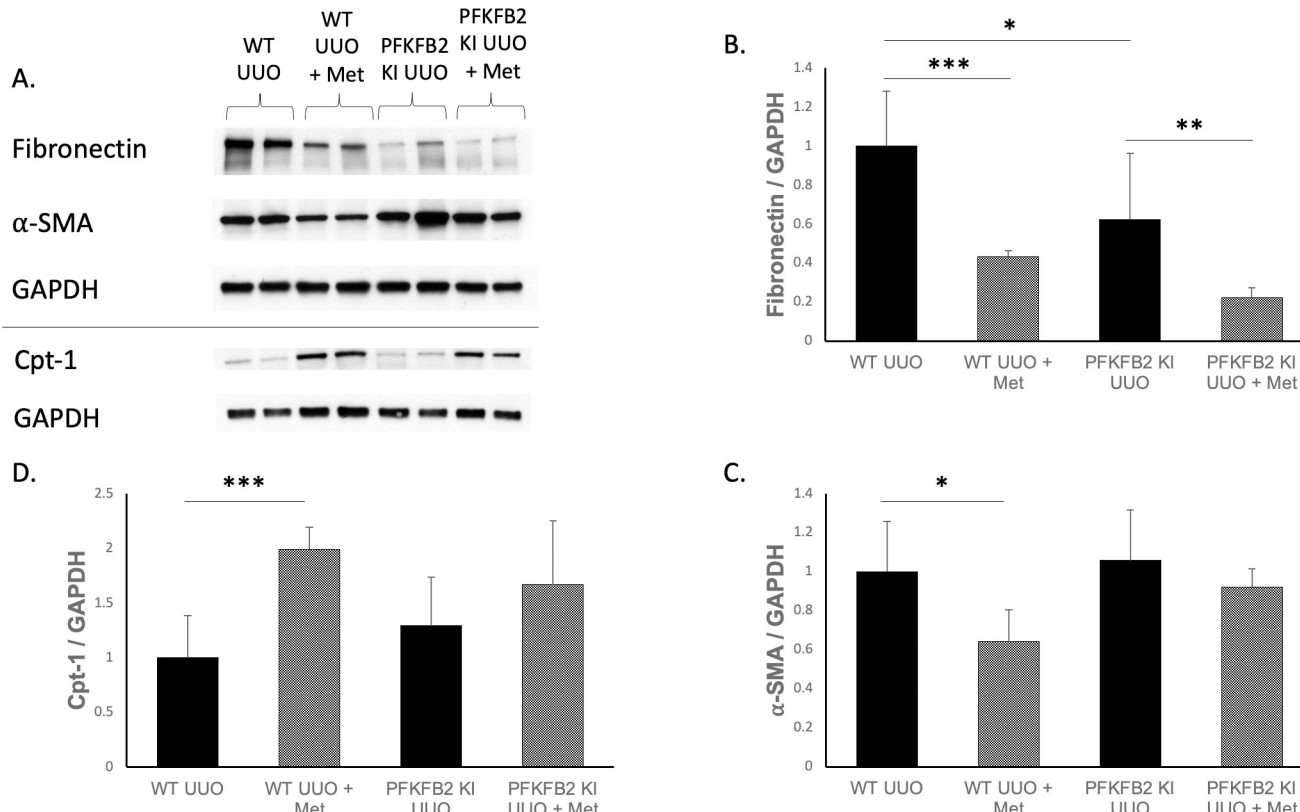

**Fig 2.** Western blot markers of fibrosis and metabolism following UUO in WT and PFKFB2 KI ± metformin kidneys (A-D). Analysis showed significantly reduced expression of fibronectin and α-SMA in WT UUO metformin kidneys versus controls (**A-C** ***p<0.001 and *p = 0.01 respectively). PFKFB2 KI UUO controls had reduced fibronectin (**A, B** *p = 0.01) but not α-SMA compared to WT (**A, C**). PFKFB2 KI UUO metformin had reduced fibronectin compared to PFKFB2 KI controls (**A, B** **p = 0.009) with a trend towards the same in α-SMA (**A, C**). Expression of Cpt-1 was increased in WT UUO + metformin kidneys compared to WT UUO counterparts (**A, D** ***p = 0.007). Mean + SD.

increase in expression of phosphorylated ACC-Ser[79] seen in both WT and PFKFB2 KI cells (Fig 1A and 1C). Expression of ACC-Ser[79] was reduced in the PFKFB2 KI TEC's compared to their WT counterparts (Fig 1A and 1C). We note that in this analysis, phosphorylated ACC and PFKFB2 are corrected for GAPDH rather than total ACC and PFKFB2, therefore, there is uncertainty as to whether the changes observed here are entirely explained by a change in the relative phosphorylation state, or whether there is also a contribution from a change in overall ACC or PFKFB2 expression.

## Renal fibrosis in WT and PFKFB2 KI mice

WT and PFKFB2 KI mice underwent UUO and their kidneys were analysed by Western blot (Fig 2), histology (Fig 3) and RT-PCR (Fig 4) for markers of fibrosis. We have previously reported that baseline fibrosis is not different in control PFKFB2 KI mice [4], therefore, sham UUO mice were not included in this study. PFKFB2 KI UUO control kidneys had reduced expression of fibronectin compared to their WT counterparts (Fig 2B) but this was not seen for α-SMA (Fig 2C). Expression of mRNA for α-SMA was reduced in PFKFB2 KI UUO control kidneys compared to their WT counterparts (Fig 4A) but this was not seen for fibronectin (Fig 4B). Cpt-1 expression was also increased in WT UUO + metformin kidneys compared to their WT counterparts (Fig 2D) with trends towards similar in PFKFB2 kidneys, suggestive of increased Cpt-1 activity and fatty acid oxidation with the addition of metformin.

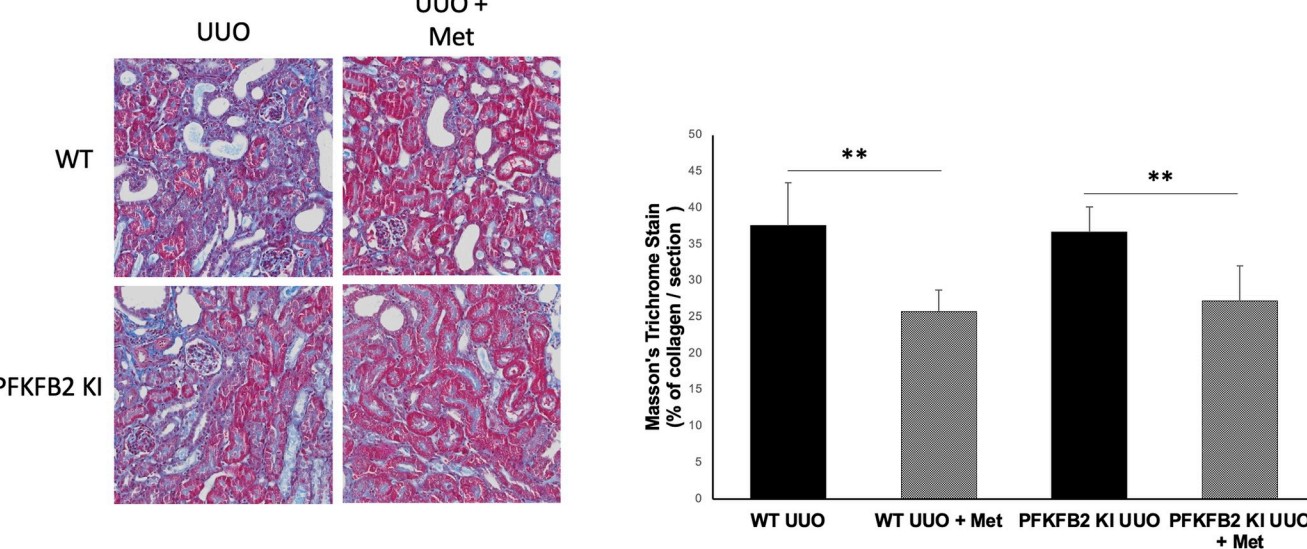

**Fig 3.** Masson's trichrome staining of collagen deposition as a marker of fibrosis for WT and PFKFB2 KI UUO kidneys ± metformin showing representative sections (A) with corresponding quantification (B). Mice treated with metformin had less evidence of collagen deposition per section compared to their untreated counterparts for both WT (**p = 0.0025) and PFKFB2 KI mice (**p = 0.0094). There was no significant difference between WT and PFKFB2 groups. Quantification was performed from a set of images that provided coverage of the whole kidney cortex. Mean + SD.

Collagen 1α and collagen 3α chain mRNA, products of the COL1A1 and COL1A3 genes respectively, were not significantly different between WT UUO and PFKFB2 KI UUO control samples (Fig 4C and 4D).

Taken together, we note that our observations using a variety of methods, including Masson Trichrome histology, Western blot (fibronectin and α-SMA), and RT-PCR (fibronectin, α-SMA, collagen 1 and 3), indicate that metformin protects against fibrosis in the UUO model, and that this effect is not altered in the PFKFB2 KI mice.

### Effects of metformin in WT and PFKFB2 KI mice

Following metformin treatment, expression of fibronectin and α-SMA protein by Western blot was reduced in WT UUO kidneys (Fig 2A–2C) as well as PFKFB2 KI UUO kidneys for fibronectin (Fig 2A and 2B). When analysing the kidneys for mRNA expression of the same markers via RT-PCR (Fig 4A and 4B), α-SMA and fibronectin were significantly reduced in WT UUO kidneys with the addition of metformin. α-SMA was reduced with metformin in the PFKFB2 KI UUO kidneys (Fig 4A) with a trend towards the same in fibronectin (Fig 4B).

The degree of fibrosis was confirmed using a Masson's trichrome stain for collagen in the tissues (Fig 3). Analysis demonstrated decreased collagen accumulation per section examined for both WT and PFKFB2 KI kidneys with the addition of metformin.

Collagen 1α and collagen 3α showed similar trends with reduced expression in mice treated with metformin, although only the results for collagen 3α were significant (Fig 4C and 4D).

The reduction in macrophage numbers that has been reported with metformin in models of renal fibrosis was seen in both WT and PFKFB2 KI UUO kidneys (Fig 5A) and TNF-α as a marker of inflammation was also similarly reduced (Fig 5B). This is despite an increase in monocyte chemoattractant protein-1 (MCP-1) expression in PFKFB2 KI UUO + metformin kidneys compared to PFKFB2 KI UUO controls and WT UUO + metformin counterparts (S1B Fig). Interleukin 1 and interleukin 6 were also reduced in WT mice treated with

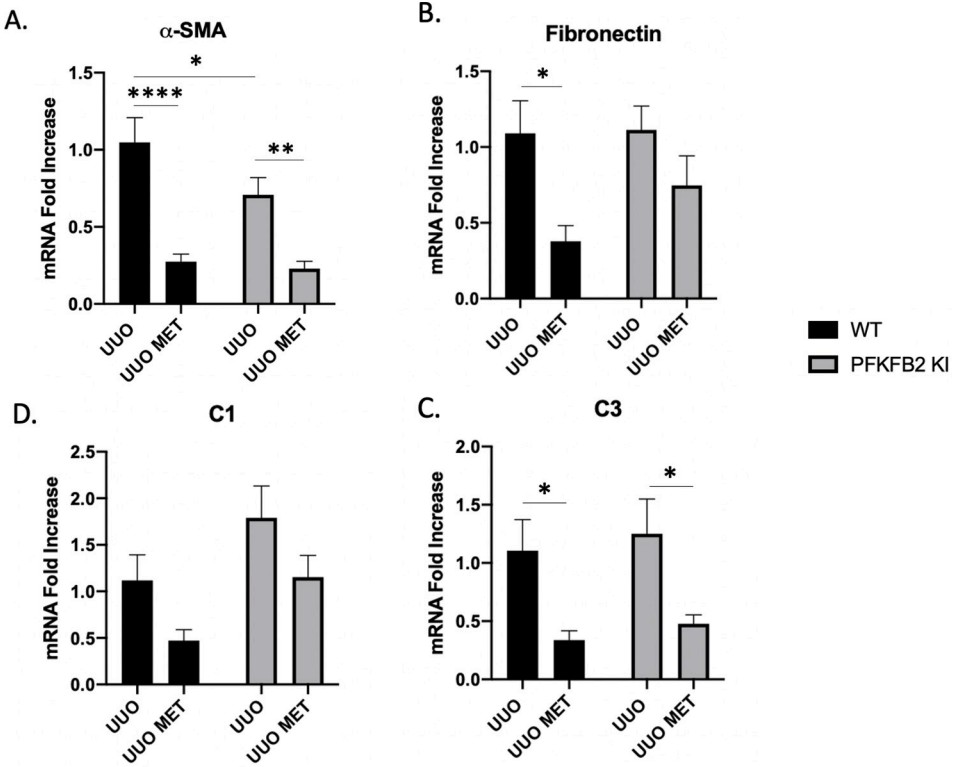

**Fig 4.** mRNA expression of markers of fibrosis for WT and PFKFB2 KI UUO kidneys ± metformin measured by RT-PCR (A-D). α-smooth muscle actin (α-SMA) was significantly lower with the addition of metformin for WT and PFKFB2 KI kidneys (**A** ****p<0.001 and **p = 0.002 respectively). Similarly, mRNA expression of fibronectin was significantly less in WT UUO kidneys treated with metformin compared to control UUO comparisons (**B** *p = 0.02) with a trend towards similar effect in PFKFB2 KI kidneys. PFKFB2 KI UUO control kidneys had significantly less expression of α-SMA compared to their WT counterparts (**A** *p = 0.04). mRNA expression of collagen 3 (C3) was significantly reduced with the addition of metformin (**C**, p<0.05) with trends towards a similar pattern in Collagen 1 (C1) (**D**) Mean + SD.

metformin and there was a trend for PFKFB2 KI mice (Fig 5C and 5D). There was no significant difference in Sirtuin 3 mRNA expression between groups (S1A Fig).

Considering the lack of an effect of mutation of the regulatory phosphorylation sites of PFKB2 on the anti-fibrotic effect of metformin, we considered the possibility that this lack of effect might be explained by an effect of metformin on the expression of the inducible PFKFB3 isoform. Interestingly, expression of total PFKFB3 was increased in UUO mice treated with metformin for both WT and PFKFB2 KI groups (Fig 6).

## Measurement of other rate-limiting steps in glycolysis

To determine whether other major rate-limiting steps were affected by the mutation of PFKFB2 phosphorylation sites or the addition of metformin, whole kidney lysates from mice were analyzed via Western blot for the expression of key enzymes in the glycolytic pathway (Fig 7). The most prevalent isoforms in the kidney for each of these rate-limiting enzymes was selected for analysis. Expression of Hexokinase-1 and PFK1 were not affected by the knock-in mutation nor stimulation by metformin (Fig 7B and 7C). PKM2 expression decreased with the addition of metformin in both WT and PFKFB2 KI samples (Fig 7D).

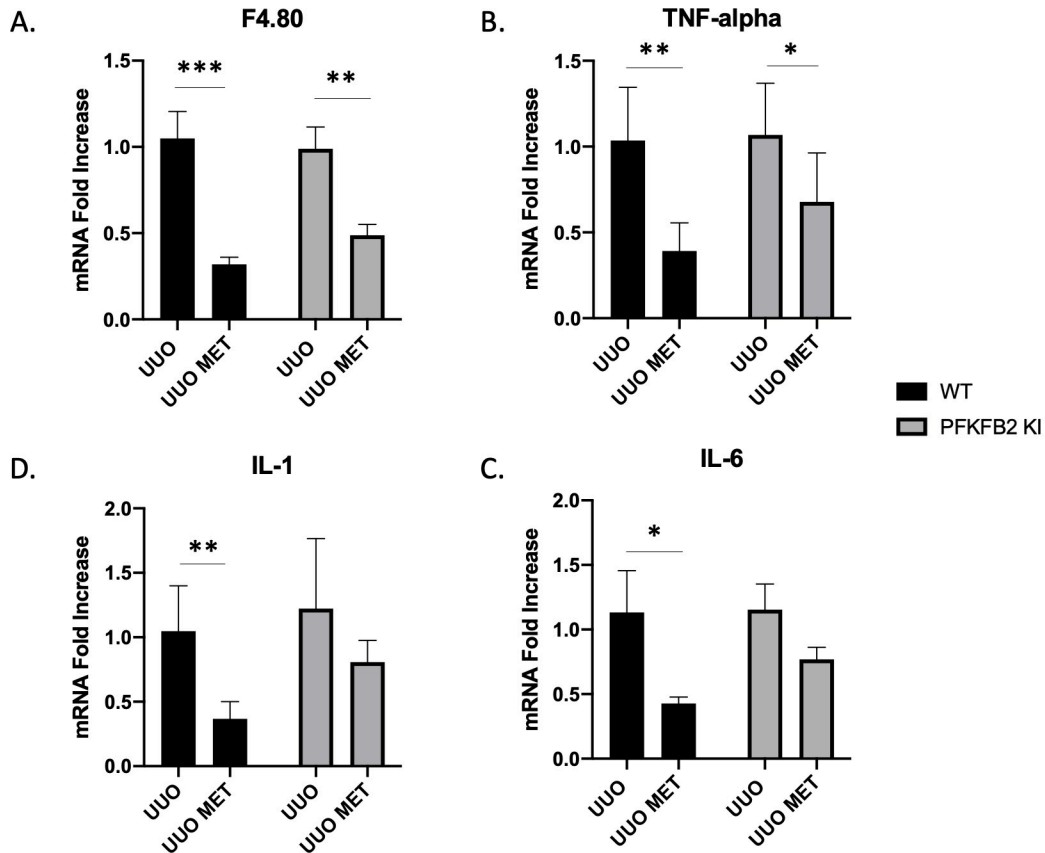

**Fig 5.** Measurement of markers of inflammation via RT-PCR in WT and PFKFB2 KI kidneys subjected to UUO ± metformin (A-D). mRNA expression of F4.80, TNF-α, Interleukin-1 (IL-1) and Interleukin-6 (IL-6) was significantly reduced with the addition of metformin in wild-type mice (**A-D** ***p = 0.0001, **p = 0.0016, *p = 0.0230, **p = 0.0008 respectively). In the PFKFB2 KI mice, there was decreased mRNA expression of F4.80 and TNF-α with the addition of metformin (**A, B** **p = 0.0024, *p = 0.0277 respectively). There was no significance difference between WT and PFKFB2 KI mouse groups in any of these parameters measured. Mean + SD.

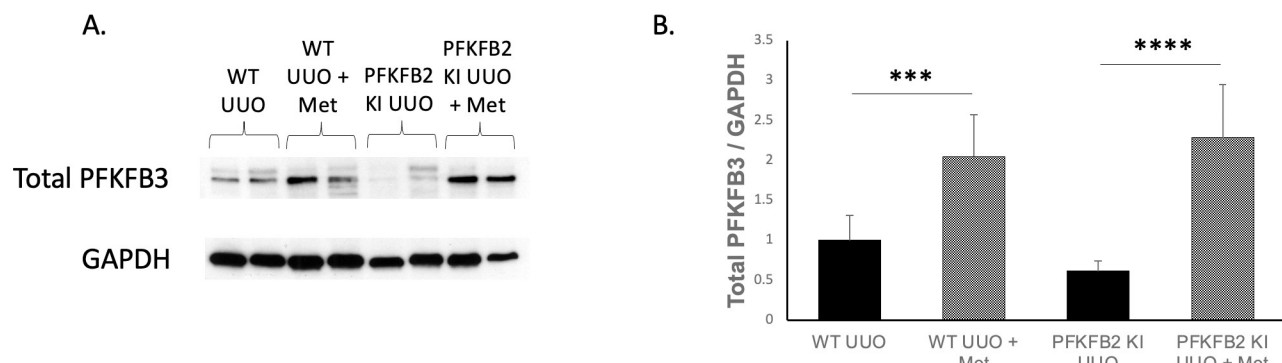

**Fig 6. Measurement of Western blot expression of total PFKFB3 in whole kidneys for WT and PFKFB2 KI UUO ± metformin.** Expression of Total PFKFB3 was increased in mice treated with metformin for both WT and PFKFB2 KI groups (**A, B** ***p = 0.0009, ****p<0.0001 respectively). Mean + SD.

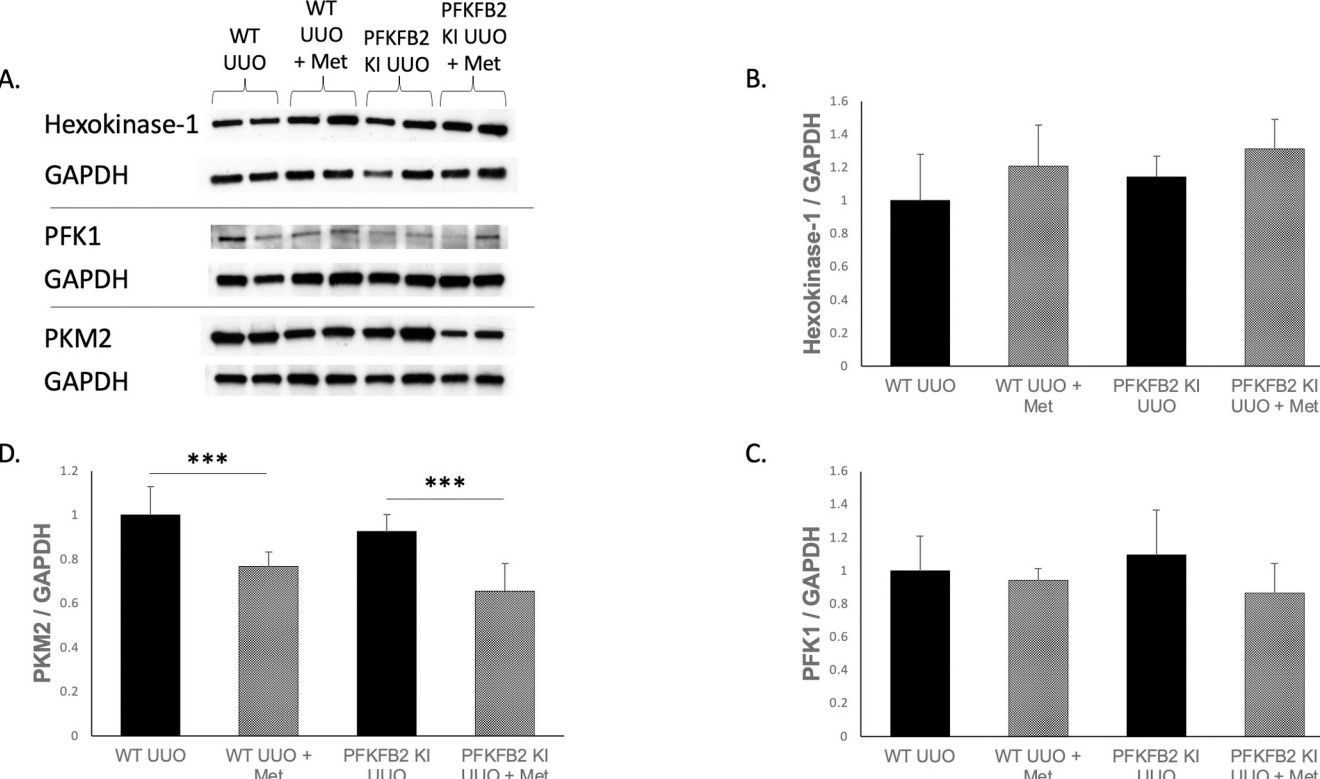

**Fig 7.** Measurement of the other rate-limiting steps in glycolysis via Western blot for WT and PFKFB2 KI UUO ± metformin kidneys (A-D). The most prevalent form of each of these enzymes in the kidney was selected for analysis. Expression of hexokinase-1 and PFK-1 was not significantly altered by genotype or presence of metformin (**B, C**). PKM2 expression was significantly reduced in WT and PFKFB2 KI kidneys with the addition of metformin (**D** ***p = 0.001 and p<0.001 respectively). Mean + SD.

## Discussion

In this study we demonstrated that metformin continues to have its protective effects in the UUO model of renal fibrosis in mice with a mutation of a key regulatory control point in glycolysis. When phosphorylated by protein kinases, PFKFB2 increases synthesis of fructose-2,6-bisphosphate, a strong activator of PFK1, the major control point in glycolysis [13]. Mutation of the two phosphorylation sites in its C-terminus reduces the increase in glycolysis seen after stimulation of cells with extracellular glucose [5]. Despite this change, metformin continued to reduce fibrosis in the UUO model. The PFKFB2 isoform was selected for investigation in this study because previous studies have found that it is the most prevalent isoform in the kidney [6]. PFKFB1 is not present in the kidney in detectable amounts, being most predominant in the heart, skeletal muscle and white adipose tissue. PFKFB4, the isoform of PFKFB specific to the testis only, is present in low levels in the kidney but is not regulated by AMPK, so is very unlikely to contribute to the action of metformin [5, 6].

Interestingly, the expression of the PFKFB3 isoform was increased in mice treated with metformin for both WT and PFKFB2 KI UUO groups, which has not previously been described. PFKFB3 is expressed predominantly in brain and placental tissue, and comprises a constitutive as well as an inducible isoform. Expression of the inducible isoform is thought to be low in adult tissues, high in tumour cell lines and increased by pro-inflammatory stimuli [14, 15]. For instance, expression has been shown to be increased in monocytes exposed to lipopolysaccharide as well as in response to hypoxia via the HIF-1 pathway, in addition to

stimuli such as progestins, insulin and protein kinase C [15–17]. Whilst AMPK-mediated phosphorylation of PFKFB3 is described in the literature, the mechanism of upregulation of total PFKFB3 levels in response to metformin is unknown. The potential effect of this upregulation might be to increase glycolysis, although downstream changes to suggest an increase in glycolysis were not seen; PFK-1 expression was unchanged and PKM2 expression was reduced with the addition of metformin as discussed below. Hence, the significance of this novel effect of metformin on PFKFB3 expression remains uncertain. It is possible that the observed increase in PFKFB3 might be a contributing reason to the lack of effect of mutation of the pFKFB2 phosphosites on the protective action of metformin observed in this study, although this speculation is unproven at the present time.

The protective effects of metformin were demonstrated in multiple markers of fibrosis via Western blot, histology and RT-PCR. The ongoing benefits that were seen in the PFKFB2 KI mice evidently reflect non-PFKFB2 mediated effects of metformin. These other effects of metformin could either be AMPK-mediated or AMPK-independent. Regarding AMPK-mediated effects, these may include metformin causing AMPK-mediated phosphorylation of ACC and consequent increased fatty acid oxidation, as this pathway was unaltered in these mice. This is evidenced by the ongoing ACC-Ser[79] activation that was demonstrated with the addition of metformin in the PFKFB2 KI cells. A limitation of our study is that we do not have data on AMPK activity in the kidneys of the metformin treated mice, which was not possible to analyse as we did not employ a free-clamp methodology for harvesting the kidney tissue. Regarding other potential mechanisms of metformin, we noted that Cpt-1 and PFKFB3 expression were increased via Western blot in WT UUO + metformin kidneys with similar changes in the PFKFB2 KI UUO + metformin mice. Given that metformin still displays a protective effect in this PFKFB2 KI model, it would suggest that the PFK pathway and up-regulation of glycolysis for ATP generation is not essential to the anti-fibrotic effects of metformin, which appear to be primarily through fatty acid oxidation.

One unexpected result was reduced expression of the pyruvate kinase isoform PKM2 in both WT and PFKFB2 KI mice receiving metformin compared to those that did not. Previous work has shown increased PKM2 expression in UUO models compared to sham controls for both WT and PFKFB2 KI kidneys [4]. PKM2 expression decreased in UUO mice treated with metformin relative to untreated mice. This suggests that metformin reduced glycolysis, independent of its action on PFKFB2, in mice with fibrosis. This may represent an unexpected and novel action of metformin in disease. However, further studies are required to demonstrate a change in glycolysis. It is also worth noting that PKM2 is active in its tetrameric form, rather than the dimeric form. A limitation of our data is that Western blots are generally unable to distinguish the tetrameric from the dimeric form, so it is unclear whether the reduction in total PKM2 protein represents a change in the number of tetramers [18]. An additional limitation of these data is that glycolysis was not directly measured, which would require either *ex vivo* studies, or measurement of tissue metabolite levels, which were not available for this study.

In summary, this study shows that metformin continues to have the effect of reducing renal fibrosis in a UUO model, despite the absence of one of the major regulatory mechanisms in control of glycolysis. This indicates that regulation of glycolysis does not play a significant role in mediating the anti-fibrotic effect of metformin.

## Supporting information

**S1 Fig. Measurement of mRNA expression via RT-PCR of other markers for WT and PFKFB2 KI UUO ± metformin kidneys (A, B).** There was no significant difference in

expression of Sirtuin 3 between groups (**A**). Expression of monocyte chemoattractant protein-1 (MCP-1) was increased in PFKFB2 KI UUO + metformin kidneys compared to PFKFB2 KI UUO controls (**B** *p = 0.0472) and WT UUO + metformin comparators (**B** **p = 0.0014). Mean + SD.
(TIF)

**S1 Raw images.**
(PDF)

## Acknowledgments

We gratefully acknowledge the laboratory of Prof Bruce Kemp at St. Vincent's Institute of Medical Research who maintained the PFKFB2 KI transgenic line for a period. Parts of this study were presented at the American Society of Nephrology Annual Meeting in 2021.

## Author Contributions

**Conceptualization:** Geoff Harley, Marina Katerelos, Peter F. Mount, David A. Power.

**Data curation:** Geoff Harley, Marina Katerelos, Kurt Gleich, Mardiana Lee.

**Formal analysis:** Geoff Harley, Mardiana Lee.

**Funding acquisition:** David A. Power.

**Investigation:** Geoff Harley, Kurt Gleich, Peter F. Mount, David A. Power.

**Methodology:** Geoff Harley, Marina Katerelos, Peter F. Mount, David A. Power.

**Supervision:** Marina Katerelos, Peter F. Mount, David A. Power.

**Validation:** Mardiana Lee.

**Writing – original draft:** Geoff Harley.

**Writing – review & editing:** Geoff Harley, Peter F. Mount, David A. Power.

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
