## [Decision Letter · Decision Letter 0]

19 Jan 2022

PONE-D-21-37485Mutation of regulatory phosphorylation sites in PFKFB2 does not affect the anti-fibrotic effect of metformin in the kidneyPLOS ONE

Dear Dr. Harley,

Thank you for submitting your manuscript to PLOS ONE. After careful consideration, we feel that it has merit but does not fully meet PLOS ONE’s publication criteria as it currently stands. Therefore, we invite you to submit a revised version of the manuscript that addresses the points raised during the review process.

Your manuscript was reviewed by two expert investigators and both of them gave valuable feedback. Please address those comments as appropriate.

We look forward to receiving your revised manuscript.

Kind regards,

Partha Mukhopadhyay, Ph.D.

Academic Editor

PLOS ONE

Journal Requirements:

(G.H. was supported by a postgraduate scholarship from the University of Melbourne. We acknowledge the laboratory of Prof Bruce Kemp at St. Vincent’s Institute of Medical Research who maintained the PFKFB2 KI transgenic line for a period. Parts of this study were presented at the American Society of Nephrology Annual Meeting in 2021.)

4.Thank you for stating the following in the Acknowledgments Section of your manuscript: 

(G.H. was supported by a postgraduate scholarship from the University of Melbourne. We acknowledge the laboratory of Prof Bruce Kemp at St. Vincent’s Institute of Medical Research who maintained the PFKFB2 KI transgenic line for a period. Parts of this study were presented at the American Society of Nephrology Annual Meeting in 2021.)

(D.P. received a National Health and Medical Research Council (NHMRC) grant 

www.nhmrc.gov.au

D.P. was involved in study design, decision to publish and preparation of the manuscript)

Reviewers' comments:

Reviewer's Responses to Questions

**Comments to the Author**

1. Is the manuscript technically sound, and do the data support the conclusions?

Reviewer #1: Yes

Reviewer #2: Partly

2. Has the statistical analysis been performed appropriately and rigorously? 

Reviewer #1: Yes

Reviewer #2: Yes

3. Have the authors made all data underlying the findings in their manuscript fully available?

Reviewer #1: Yes

Reviewer #2: Yes

4. Is the manuscript presented in an intelligible fashion and written in standard English?

Reviewer #1: Yes

Reviewer #2: Yes

5. Review Comments to the Author

Reviewer #1: In the present manuscript Harley et al. reported the anti-fibrotic effect of metformin in vivo in a mouse model of renal fibrosis induced by unilateral ureteral occlusion (UUO). This anti-fibrotic effect of metformin was independent from the rate of glycolysis. The authors used a 7-day UUO model in both wild-type and genetically modified mice (PFKFB2 KI) lacking a key regulatory point of glycolysis (mutated phosphorylation sites of phosphofructokinase-2) resulting in a reduced ability to increase glycolysis in the kidneys. Despite this change, metformin continued to reduce fibrosis in this UUO model.

The goals are clear, the manuscript is well-written and readable, however, there are some typos that should be corrected in the final version.

Nevertheless, there are major comments need to be addressed:

1. The authors used assessed the severity of renal fibrosis by assessing some key markers of fibrosis including a-SMA or fibronectin at protein or gene expression levels (Figs. 2-3). However, showing representative micrographs (e.g.: Masson’s trichrome or Sirius red staining of the affected kidneys would be a nice addition and improve the quality of the paper.

2. Please put in data from sham operated animals shown in a separate, supplementary figure set.

3. Based on the relevant literature, The mitochondrial deacetylase sirtuin 3 (Sirt3) is involved the stress response activating mitochondrial enzymes involved in fatty acid oxidation, amino acid metabolism, electron transport chain activity, etc. Furthermore, Sirt3 deactivation is a key player in renal fibrosis resulting in epithelial-mesenchymal transition. Did the authors assess the role of Sirt3 in this study? How would a dysregulated glycolysis affect the activity of Sirt3 in the presence or absence of metformin in this UUO model?

4. Did the authors see any changes in parameters describing kidney function (e.g.: NGAL levels reflecting injury or BUN, glomerular filtration rate, etc.) following metformin treatment of either wild-type or PFKFB2 KI mice?

5. Also, please provide information on the levels of other inflammatory cell markers (e.g.: using Ly6C, Ly6G, CD45 immunostainings or RT-PCR) in the injured kidneys.

6. Similarly, tissue levels of other (more conventional) cytokines should be also measured (e.g.: TNFa, IL-1b, IL-17, IL-18, IL-33, MCP-1, MIP-1a).

7. Based on Fig 5, both wild-type and PFKFB2 KI mice represented a reduced expression of pyruvate kinase (PKM2). Therefore, measuring the levels of an upstream metabolite (e.g.:2,3 bis-phosphoglycerate) would determine if glycolysis is downregulated in renal fibrosis.

Reviewer #2: In the present study “Mutation of regulatory phosphorylation sites in PFKFB2 does not affect the anti-fibrotic effect of metformin in the kidney” the authors analyzed the effect of PFKB2 on kidney fibrosis and the contribution to the protective function of metformin. Here are some major concerns to be addressed:

1. The authors need to examine the expression level of PFKFB1, 3 and 4, as well as their phosphorylation levels in the kidneys.

2. The authors need to show the metabolic features of the PFKFB2 KI mice with and without metformin, including glucose, amino acid and fatty acids metabolism.

3. A morphological study should be performed on the kidney samples for fibrosis, such as Masson Trichrome staining or Sirius Rid Staining. A hydroxyproline quantification should also be performed.

4. The authors should show the baseline level of all the parameters in this study by examining and exhibiting the unchallenged WT and PFKFB2 KI mice.

5. AMPK phosphorylation should be examined in this study.

6. To define a negative result, the authors need to also establish a positive control in the same condition. Because the authors’ previous study to determine the function of AMPK-ACC pathway in metformin action on kidney fibrosis is performed on a different kidney fibrosis model, it cannot be directly used as a positive control. The authors need to use the ACC KI mice in the current model to establish a positive control.

7. Blood creatinine and BUN should be tested

8. Why would metformin not increase the phosphorylation of PFKFB2 in wild type mice?

6. PLOS authors have the option to publish the peer review history of their article (what does this mean?). If published, this will include your full peer review and any attached files.

Reviewer #1: No

Reviewer #2: No

---

## [Author Response · Author response to Decision Letter 0]

26 Jul 2022

Reviewer #1: In the present manuscript Harley et al. reported the anti-fibrotic effect of metformin in vivo in a mouse model of renal fibrosis induced by unilateral ureteral occlusion (UUO). This anti-fibrotic effect of metformin was independent from the rate of glycolysis. The authors used a 7-day UUO model in both wild-type and genetically modified mice (PFKFB2 KI) lacking a key regulatory point of glycolysis (mutated phosphorylation sites of phosphofructokinase-2) resulting in a reduced ability to increase glycolysis in the kidneys. Despite this change, metformin continued to reduce fibrosis in this UUO model.

The goals are clear, the manuscript is well-written and readable, however, there are some typos that should be corrected in the final version.

Nevertheless, there are major comments need to be addressed:

1. The authors used assessed the severity of renal fibrosis by assessing some key markers of fibrosis including a-SMA or fibronectin at protein or gene expression levels (Figs. 2-3). However, showing representative micrographs (e.g.: Masson’s trichrome or Sirius red staining of the affected kidneys would be a nice addition and improve the quality of the paper.

We have undertaken additional histology work to confirm these findings using quantitative analysis of a Masson’s trichrome stain and added this as Figure 3. In brief, it confirmed the anti-fibrotic effects of metformin in both WT and PFKFB2 KI UUO groups, similar to the existing Western blot and RT-PCR data. We have updated the results section and abstract to include this additional data.

2. Please put in data from sham operated animals shown in a separate, supplementary figure set.

Our group has previously published data regarding the comparison between sham and UUO-operated mice using an identical UUO technique for both WT and PFKFB2 KI mice (Lee et al., 2020) so this was not repeated as part of this paper. We have now explained this point in the results section of the revised manuscript.

3. Based on the relevant literature, The mitochondrial deacetylase sirtuin 3 (Sirt3) is involved the stress response activating mitochondrial enzymes involved in fatty acid oxidation, amino acid metabolism, electron transport chain activity, etc. Furthermore, Sirt3 deactivation is a key player in renal fibrosis resulting in epithelial-mesenchymal transition. Did the authors assess the role of Sirt3 in this study? How would a dysregulated glycolysis affect the activity of Sirt3 in the presence or absence of metformin in this UUO model?

We have measured Sirtuin 3 mRNA expression via RT-PCR and not found it to be significantly different between groups. This data has been added as part of Supplementary Figure 2 and explained in the results section.

4. Did the authors see any changes in parameters describing kidney function (e.g.: NGAL levels reflecting injury or BUN, glomerular filtration rate, etc.) following metformin treatment of either wild-type or PFKFB2 KI mice?

These parameters were not specifically measured. Since the UUO model leaves the contralateral kidney left intact, in previous studies, we have not found whole body markers of renal function to be useful outcome measures.

5. Also, please provide information on the levels of other inflammatory cell markers (e.g.: using Ly6C, Ly6G, CD45 immunostainings or RT-PCR) in the injured kidneys.

These markers were not specifically examined, however, additional RT-PCR analysis of other inflammatory markers has been added to the manuscript as part of the response to the next query.

6. Similarly, tissue levels of other (more conventional) cytokines should be also measured (e.g.: TNFa, IL-1b, IL-17, IL-18, IL-33, MCP-1, MIP-1a).

Additional inflammatory markers have been measured via RT-PCR analysis. These include TNF-alpha, MCP-1 and IL-1. They have been included as part of figure 5 and supplementary figure 2. Similar to the fibrosis markers, the inflammation markers F4.80, TNF-alpha, IL-1 and IL-6 were reduced with metformin and not significantly different between WT and PFKFB2 KI mice. Explanation of these data has been added to the text of the results.

7. Based on Fig 5, both wild-type and PFKFB2 KI mice represented a reduced expression of pyruvate kinase (PKM2). Therefore, measuring the levels of an upstream metabolite (e.g.:2,3 bis-phosphoglycerate) would determine if glycolysis is downregulated in renal fibrosis.

As mentioned in our discussion, this decrease in PKM2 expression may represent downstream reduced glycolysis as an unexpected action of metformin in this model, however, further studies would be needed to confirm this finding, particularly given the issue of dimers and tetramers of PKM2 being relevant to its overall activity. Assuming this change in Western blot expression correlated with overall activity, further upstream metabolites weren’t specifically examined but would be of interest to explore this effect. However, the manuscript isn’t seeking to make a definitive conclusion about metformin’s effects on the rate of glycolysis, only to show that despite absence of one of the major regulatory mechanisms in control of glycolysis, metformin continues to have significant anti-fibrotic effects. We have added explanation in the discussion (second last paragraph), that a limitation of the study is that glycolylysis was not directly measured.

Reviewer #2: In the present study “Mutation of regulatory phosphorylation sites in PFKFB2 does not affect the anti-fibrotic effect of metformin in the kidney” the authors analyzed the effect of PFKB2 on kidney fibrosis and the contribution to the protective function of metformin. Here are some major concerns to be addressed:

1. The authors need to examine the expression level of PFKFB1, 3 and 4, as well as their phosphorylation levels in the kidneys.

As established by the work of Minchenko et al. (Minchenko et al., 2003) and others, PFKFB 1, 3 and 4 exist at lower expression levels in mouse kidney tissue compared with PFKFB2. This was the reason we selected PFKFB2 as the most prevalent isoform. PFKFB1 is not present in the kidney in detectable amounts. PFKFB3 and 4 are present, but we have not attempted further study in view of our focus on the dominant isoform, PFKFB2. Additional information has been added to the first paragraph of the discussion to better explain this point.

2. The authors need to show the metabolic features of the PFKFB2 KI mice with and without metformin, including glucose, amino acid and fatty acids metabolism.

As stated in the manuscript, in the methods section under the “Generation of PFKFB2 KI mice” heading, we have previously published the observed phenotype of the PFKFB2 KI mice (Lee et al., 2020). There was no difference in plasma glucose or mouse weight between PFKFB2 KI mice and controls, however, the PFKFB2 KI kidneys were smaller and plasma urea was significantly less. Furthermore, cultured tubular epithelial cells from PFKFB2 KI mice have impaired glycolysis when analysed on the Seahorse analyser (Lee et al., 2020). We have added this information to this section of the methods.

In this revised manuscript we have added Western blot expression of Cpt-1 (Fig 2D) as a marker of fatty acid metabolism, which interestingly was improved with metformin in WT, and followed a similar trend in the PFKFB2 KI mice. This additional data is not explained in the Results section under the “Renal fibrosis in WT and PFKFB2 KI mice” subheading. 

3. A morphological study should be performed on the kidney samples for fibrosis, such as Masson Trichrome staining or Sirius Rid Staining. A hydroxyproline quantification should also be performed.

Please see our response for query 1 by Reviewer #1. Masson’s trichrome staining was performed to quantify the degree of fibrosis and an additional figure has been added to our manuscript (Fig 3). These data are presented in the Results section under the “Effects of metformin in WT and PFKFB2 KI mice” subheading. Hydroxyproline quantification was not performed given these results were consistent with the pattern of Western blot and RT-PCR data presented.

4. The authors should show the baseline level of all the parameters in this study by examining and exhibiting the unchallenged WT and PFKFB2 KI mice.

As discussed above, we have previously published on the phenotype of sham-operated WT and PFKFB2 KI mice using an identical UUO experimental setup (Lee et al., 2020) so that work was not replicated in this paper.

5. AMPK phosphorylation should be examined in this study.

This has been added as part of Supplementary Figure 1. We observed that expression of phosphorylated Thr172, the phosphorylation site on AMPK, was increased in PFKFB2 KI UUO kidneys compared to WT counterparts, a finding of uncertain significance. This additional data has been outlined in the Results section, in the second paragraph of the “Renal fibrosis in WT and PFKFB2 KI mice” section.

6. To define a negative result, the authors need to also establish a positive control in the same condition. Because the authors’ previous study to determine the function of AMPK-ACC pathway in metformin action on kidney fibrosis is performed on a different kidney fibrosis model, it cannot be directly used as a positive control. The authors need to use the ACC KI mice in the current model to establish a positive control.

Our group has used a similar experimental setup in a previous published work in JASN (Lee et al., 2018). In this work the same experimental setup was used – three days of metformin in the drinking water of the mice then a folate nephropathy model and assessment of the degree of kidney fibrosis seven days later. This was done in both WT and ACC1/2 KI mice as a positive control of the anti-fibrotic effects of metformin. Likewise, in this current study, we demonstrated increased ACC-Ser79 activation with metformin even in PFKFB2 KI mice, suggesting ongoing effects on the ACC pathway and leading to our hypothesized explanation of the ongoing protective effects of metformin. 

The protective effects of metformin in UUO models of fibrosis have been well established in the past; for instance Cavaglieri et al. used a UUO model in WT C57Bl/6 mice with metformin given one day prior to surgery and the outcome assessed seven days later compared to sham controls (Cavaglieri et al., 2015). They demonstrated a significant protective of metformin based on Western blot and histological measures of fibrosis. The paper by Shen et al. is another example of this (Shen et al., 2016).

7. Blood creatinine and BUN should be tested

These parameters were not specifically measured. Since this is a UUO model with the contralateral kidney left intact, whole body markers of renal function are not useful outcome measures.

8. Why would metformin not increase the phosphorylation of PFKFB2 in wild type mice?

As discussed in the first paragraph of the results section, PFKFB2-Ser483 is not a phosphosite for AMPK, so metformin should have little effect on its phosphorylation. PFKFB2-Ser466 is the active phosphorylation site for AMPK, however, due to sequence homology it cannot be distinguished from a similar site in PFKFB3. Hence, demonstration of increased Western blot expression would be unable to distinguish between increased phosphorylation on PFKFB2 or PFKFB3 so would not be valid.

Amended funding statement

G. H. was supported by a postgraduate scholarship from the University of Melbourne. D.P. received a National Health and Medical Research Council (NHMRC) grant 

www.nhmrc.gov.au. There was no additional external funding received for this study.

References:

Cavaglieri, R., Day, R., Feliers, D., & Abboud, H. (2015). Metformin prevents renal interstitial fibrosis in mice with unilateral ureteral obstruction. Molecular and Cellular Endocrinology, 412, 116-122. 

Lee, M., Harley, G., Katerelos, M., Gleich, K., Sullivan, M., Laskowski, A., Coughlan, M., Fraser, S., Mount, P., & Power, D. (2020). Mutation of regulatory phosphorylation sites in PFKFB2 worsens renal fibrosis. Scientific Reports, 10, 14531. 

Lee, M., Katerelos, M., Gleich, K., Galic, S., Kemp, B., Mount, P., & Power, D. (2018). Phosphorylation of Acetyl-CoA Carboxylase by AMPK Reduces Renal Fibrosis and Is Essential for the Anti-Fibrotic Effect of Metformin. JASN, 29(9), 2326-2336. 

Minchenko, O., Opentanova, I., & Caro, J. (2003). Hypoxic regulation of the 6-phosphofructo-2-kinase/fructose-2,6-bisphosphatase gene family (PFKFB-1-4) expression in vivo. FEBS Letters, 554(3), 264-270. 

Shen, Y., Miao, N., Xu, J., Gan, X., Xu, D., Zhou, L., Xue, H., Zhang, W., & Lu, L. (2016). Metformin Prevents Renal Fibrosis in Mice with Unilateral Ureteral Obstruction and Inhibits Ang II-Induced ECM Production in Renal Fibroblasts. Int. J. Mol. Sci, 17, 146.

---

## [Decision Letter · Decision Letter 1]

21 Sep 2022

PONE-D-21-37485R1Mutation of regulatory phosphorylation sites in PFKFB2 does not affect the anti-fibrotic effect of metformin in the kidneyPLOS ONE

Dear Dr. Harley,

Thank you for submitting your manuscript to PLOS ONE. After careful consideration, we feel that it has merit but does not fully meet PLOS ONE’s publication criteria as it currently stands. Therefore, we invite you to submit a revised version of the manuscript that addresses the points raised during the review process.

We look forward to receiving your revised manuscript.

Kind regards,

Partha Mukhopadhyay, Ph.D.

Section Editor

PLOS ONE

Additional Editor Comments:

Some of you data are inconsistent and one of the reviewer has pointed out clearly. Please address those comments as earliest.

Reviewers' comments:

Reviewer's Responses to Questions

**Comments to the Author**

1. If the authors have adequately addressed your comments raised in a previous round of review and you feel that this manuscript is now acceptable for publication, you may indicate that here to bypass the “Comments to the Author” section, enter your conflict of interest statement in the “Confidential to Editor” section, and submit your "Accept" recommendation.

Reviewer #1: All comments have been addressed

Reviewer #2: (No Response)

2. Is the manuscript technically sound, and do the data support the conclusions?

Reviewer #1: Yes

Reviewer #2: Partly

3. Has the statistical analysis been performed appropriately and rigorously? 

Reviewer #1: Yes

Reviewer #2: Yes

4. Have the authors made all data underlying the findings in their manuscript fully available?

Reviewer #1: Yes

Reviewer #2: Yes

5. Is the manuscript presented in an intelligible fashion and written in standard English?

Reviewer #1: Yes

Reviewer #2: Yes

6. Review Comments to the Author

Reviewer #1: The authors have improved the paper considerably by incorporating suggestions and remarks of the reviewers and providing appropriate replies to the queries issued by the reviewers.

Reviewer #2: The current version of "Mutation of regulatory phosphorylation sites in PFKFB2 does not affect the anti-fibrotic effect of metformin in the kidney" made much improvement compared to the original version. However, some major concerns still exist:

1. The authors need to show the expression and phosphorylation of PFKFB1, 3 and 4. The present research shows a negative result, attempting to prove that the PFKFB2 function is not important. However, very possibly the other family members have increased their expression and function in compensation. In such case, the conclusion should be rewritten.

2. In Figure 3, the images show much lower fibrosis in PFKFB2 KI UUO mice than WT UUO mice. It is the mildest in fibrosis in all groups. This is inconsistent with the statistical analysis. The authors should better analyze the data and make a more solid conclusion.

3. Because the Masson’s Trichrome stain result shown in Figure 3, the authors need to perform hydroxyproline analysis.

4. In supplemental figure 1, why would not metformin increase the AMPK phosphorylation in any groups? If the authors have trouble with AMPK analysis, this data should not be shown in the study, but discussed. However, at least 2 downstream phosphorylation of AMPK should be presented. The authors already showed ACC. Another downstream of AMPK should be shown. PFKFB family members should suffice.

5. The authors need to use the total protein blotting instead of GAPDH to calculate the phosphorylation level, such as PFKFBs, ACC and AMPK.

7. PLOS authors have the option to publish the peer review history of their article (what does this mean?). If published, this will include your full peer review and any attached files.

Reviewer #1: No

Reviewer #2: No

---

## [Author Response · Author response to Decision Letter 1]

7 Dec 2022

1. The authors need to show the expression and phosphorylation of PFKFB1, 3 and 4. The present research shows a negative result, attempting to prove that the PFKFB2 function is not important. However, very possibly the other family members have increased their expression and function in compensation. In such case, the conclusion should be rewritten.

Thank you for this suggestion. In response to this we have undertaken quantification of total PFKFB3 levels by Western blot which, interestingly, showed up-regulation of PFKFB3 expression with metformin – a previously undescribed finding (see new Fig 6). The abstract has been revised to include this observation (see page 2, line 27-28). A description of the observation about increased PFKFB3 has been added to the results section (see page 11, line 281-282). Furthermore, an additional paragraph has been added to the discussion to address this important finding (see pages 13-14). The original figure 6 has been re-ordered as figure 7 to maintain linearity in the manuscript. As outlined in the discussion – PFKFB1 is not present in the kidney in detectable amounts. Likewise, PFKFB4 is present in low levels in the kidney but is not regulated by AMPK, so is unlikely to contribute to the action of metformin (see Minchenko et al. ref 6 of the manuscript). Regarding the interesting point raised here by the reviewer, we have concluded this additional paragraph with an explanation stating that “It is possible that the observed increase in PFKFB3 might be a contributing reason to the lack of effect of mutation of the PFKFB2 phosphosites on the protective action of metformin observed in this study, although this speculation is unproven at the present time”. Finally, we have added three new references to the manuscript (see references 15-17), to assist with placing this additional data about PFKFB3 in context. 

2. In Figure 3, the images show much lower fibrosis in PFKFB2 KI UUO mice than WT UUO mice. It is the mildest in fibrosis in all groups. This is inconsistent with the statistical analysis. The authors should better analyze the data and make a more solid conclusion.

Thank you to the reviewer for this useful observation. The quantitative analysis in Fig 3 is based on measurement of a set of images that provide coverage of the whole kidney cortex. The methods section has been revised to make our methodology clearer for the reader (see page 6, line 139-141). Furthermore, the legend for figure 3 has been revised to make this point clearer (see page 10, line 239-240). The reviewer has correctly identified that in the previously submitted version of the manuscript that selected representative images did not well represent our quantification based on the whole kidney cortex. To address this, we have revised figure 3, by selecting images that more accurately represent our overall quantified result (see revised Fig 3).

3. Because the Masson’s Trichrome stain result shown in Figure 3, the authors need to perform hydroxyproline analysis.

Thank you for this suggestion. We have reviewed the method for hydroxyproline analysis as well as commercial kits that are available. This analysis requires untreated frozen tissue. Unfortunately, the unilateral ureteric obstruction model yields a limited amount of tissue which we have already processed for Western Blot, histology and RT-PCR analyses. There is none left for another analysis. However, we do note that we have already quantified fibrosis by two other methods in addition to the Masson Trichrome histology, namely Western blot for fibronectin and α-SMA (see Fig 2) and RT-PCR for fibronectin, α -SMA, collagen 1 and collagen 3 (Fig 4), and the results were all consistent. Our assessment, therefore, is that our study already contains adequate quantification of kidney fibrosis. If we had capacity to add hydroxyproline analysis to our manuscript, this could provide some further confirmation, but our assessment is that it is very unlikely to alter the overall conclusion of our study. To clarify this point clearly for the readers we have added a new sentence in the results section (page 9, line 216-219) explaining that “Taken together, we note that our observations using a variety of methods, including Masson Trichrome histology, Western blot (fibronectin and α-SMA), and RT-PCR (fibronectin, α-SMA, collagen 1 and 3), indicate that metformin protects against fibrosis in the UUO model, and that this effect is not altered in the PFKFB2 KI mice”.

4. In supplemental figure 1, why would not metformin increase the AMPK phosphorylation in any groups? If the authors have trouble with AMPK analysis, this data should not be shown in the study, but discussed. However, at least 2 downstream phosphorylation of AMPK should be presented. The authors already showed ACC. Another downstream of AMPK should be shown. PFKFB family members should suffice.

Thank you for this observation. We agree with the reviewer that it is unusual that AMPK phosphorylation does not increase with metformin. We suspect this may be a methodological issue, as it is well described that AMPK phosphorylation can be influenced by the method of tissue harvesting, and we note that we did not use the freeze-clamp in situ method in this study, which is generally regarded as the gold standard for analysis of AMPK activity in tissue samples. As the reviewer has recommended, because of the uncertainty about the validity of this analysis, we have omitted the AMPK blots from the revised version. We have added explanation of this limitation of our study to the discussion (lines 362-364), where we explain that “A limitation of our study is that we do not have data on AMPK activity in the kidneys of the metformin treated mice, which was not possible to analyse as we did not employ a freeze-clamp methodology for harvesting the kidney tissue”.

Another consideration is that the sequence homology between PFKFB2 and PFKFB3 precludes the ability to quantify their phosphorylated forms in isolation to each other. The PFKFB1 and PFKFB4 isoforms are not significantly expressed in the kidney and not known to be phosphorylated by AMPK.

5. The authors need to use the total protein blotting instead of GAPDH to calculate the phosphorylation level, such as PFKFBs, ACC and AMPK.

Thank you for identifying this issue. As the reviewer has identified, in Fig 1, we corrected for GAPDH instead of the total ACC and PFKFB2. We acknowledge that there is potential inaccuracy to describe this as quantification of phosphorylation (defined as phosphor/total). As the reviewer has identified, what we have measured is the overall expression of phosphorylated protein (phosphor/GAPDH), which could be influenced by both the phosphorylation state and the total abundance of the protein. Unfortunately, we do not have left over samples to repeat the analysis shown in figure 1. To address this concern, we have substantially revised the wording of the results describing the TEC data in figure 1 (both text and figure legend, see changes to pages 7-8). Specifically, we have explained to the reader that “We note that in this analysis, phosphorylated ACC and PFKFB2 are corrected for GAPDH rather than total ACC and PFKFB2, therefore, there is uncertainty as to whether the changes observed here are entirely explained by a change in the relative phosphorylation state, or whether there is also a contribution from a change in overall ACC or PFKFB2 expression” (lines 180-184).

Regarding the AMPK blot (previous Supp Fig 1), as per point 4 above, we have removed this data from the revised version.

---

## [Decision Letter · Decision Letter 2]

10 Jan 2023

Mutation of regulatory phosphorylation sites in PFKFB2 does not affect the anti-fibrotic effect of metformin in the kidney

PONE-D-21-37485R2

Dear Dr. Harley,

We’re pleased to inform you that your manuscript has been judged scientifically suitable for publication and will be formally accepted for publication once it meets all outstanding technical requirements.

Kind regards,

Partha Mukhopadhyay, Ph.D.

Section Editor

PLOS ONE

Additional Editor Comments (optional):

Reviewers' comments:

Reviewer's Responses to Questions

**Comments to the Author**

1. If the authors have adequately addressed your comments raised in a previous round of review and you feel that this manuscript is now acceptable for publication, you may indicate that here to bypass the “Comments to the Author” section, enter your conflict of interest statement in the “Confidential to Editor” section, and submit your "Accept" recommendation.

Reviewer #1: All comments have been addressed

Reviewer #2: All comments have been addressed

2. Is the manuscript technically sound, and do the data support the conclusions?

Reviewer #1: Yes

Reviewer #2: (No Response)

3. Has the statistical analysis been performed appropriately and rigorously? 

Reviewer #1: Yes

Reviewer #2: (No Response)

4. Have the authors made all data underlying the findings in their manuscript fully available?

Reviewer #1: Yes

Reviewer #2: (No Response)

5. Is the manuscript presented in an intelligible fashion and written in standard English?

Reviewer #1: Yes

Reviewer #2: (No Response)

6. Review Comments to the Author

Reviewer #1: (No Response)

Reviewer #2: (No Response)

7. PLOS authors have the option to publish the peer review history of their article (what does this mean?). If published, this will include your full peer review and any attached files.

Reviewer #1: No

Reviewer #2: No

---

## [Editor Report · Acceptance letter]

31 Jan 2023

PONE-D-21-37485R2 

Mutation of regulatory phosphorylation sites in PFKFB2 does not affect the anti-fibrotic effect of metformin in the kidney 

Dear Dr. Harley:

I'm pleased to inform you that your manuscript has been deemed suitable for publication in PLOS ONE. Congratulations! Your manuscript is now with our production department. 

Kind regards, 

on behalf of

Dr. Partha Mukhopadhyay 

Section Editor

PLOS ONE